# A Systematic Review and Meta-Analysis of the Brief Cognitive Assessment for Multiple Sclerosis (BICAMS) International Validations

**DOI:** 10.3390/jcm12020703

**Published:** 2023-01-16

**Authors:** Hannah Potticary, Dawn Langdon

**Affiliations:** Department of Psychology, Royal Holloway, University of London, Egham, Surrey TW20 0EX, UK

**Keywords:** multiple sclerosis, Brief International Cognitive Assessment for Multiple Sclerosis, BICAMS, cognition, systematic review, meta-analysis

## Abstract

Cognitive impairment is a prevalent and debilitating symptom of multiple sclerosis (MS) but is not routinely addressed in clinical care. The Brief Cognitive Assessment for Multiple Sclerosis (BICAMS) was developed in 2012 to screen and monitor MS patients’ cognition. This systematic review and meta-analysis aimed to identify, synthesise, and critically appraise current BICAMS’ international validations. The literature search was conducted using PubMed, PsycINFO and Web of Science electronic databases in August 2022. Quantitative, peer-reviewed adult studies, which followed the BICAMS international validation protocol and were published in English, were included. The search identified a total of 203 studies, of which 26 were eligible for inclusion. These reported a total of 2833 adults with MS and 2382 healthy controls (HC). The meta-analysis showed that BICAMS identified impaired cognitive functioning in adults with MS compared to HC for all three subtests: information processing speed (g = 0.854, 95% CI = 0.765, 0.944, *p* < 0.001), immediate verbal recall (g = 0.566, 95% CI = 0.459, 0.673, *p* < 0.001) and immediate visual recall (g = 0.566, 95% CI = 0.487, 0.645, *p* < 0.001). Recruitment sites and strategies limit the generalisability of results. BICAMS is a valid and feasible international MS cognitive assessment.

## 1. Introduction

Cognition is a significant component of most neurodegenerative conditions and yet systematic, internationally valid measurement remains elusive for most. Early recognition of cognitive impairment allows for diagnosis and appropriate treatment, education, psychosocial support and engagement in shared decision-making regarding life planning, health care, involvement in research and financial matters [1]. It has been hard to meet the challenge of psychometrically sound, clinically feasible assessments, with some exceptions [2]. There is increasing recognition that the stimuli should not disadvantage any particular cultures [3]. Harmonisation of data across different national and ethnic communities needs careful consideration of cultural and linguistic variables [4]. The increasingly diverse populations within individual countries require health services to be agile and inclusive [5]. Important steps to advance cognitive measurement technology are global collaboration and a consensus, credible international validation protocol.

Multiple sclerosis (MS) is a chronic autoimmune-mediated disease of the central nervous system, involving inflammatory and degenerative processes [6]. This can produce a constellation of symptoms in the physical, psychiatric and cognitive domains. MS affects over 2.8 million people worldwide [7] and is typically diagnosed in adults aged 20 to 30 years [8]. Cognitive impairment is a prevalent and debilitating symptom of MS, affecting between 40–65% of patients [9]. It can be observed in all subtypes (Relapsing Remitting Multiple Sclerosis, RRMS; Secondary Progressive Multiple Sclerosis, SPMS; Primary Progressive Multiple Sclerosis, PPMS [10]), but severe cognitive impairment predominates in the progressive forms of the disease [11]. There are often marked deficits in information processing speed, attention, working memory and executive functioning [9]. It has a negative impact on quality of life [9], including activities of daily living [12], employment [13], disease management [14,15], personality [16] and driving safety [17]. Given the significant adverse consequences of cognitive difficulties, early identification of cognitive status is needed to facilitate optimal management and preserve quality of life in people with MS (PwMS [18]).

Cognitive impairment remains a neglected and under-diagnosed symptom of MS. The “invisibility” of cognitive difficulties has meant they are often overlooked by family members, colleagues and healthcare professionals since there is no obvious external disability [19]. At routine consultation, neurologists are poor at identifying MS-related cognitive impairment [20]. There is a growing consensus, across MS patients and professionals, that routine cognitive testing should form part of clinical practice to inform management [21]. Despite this, objective cognitive testing is rarely delivered [22,23]. Both the National Institute for Health and Care Excellence (NICE [24]) and the American Academy of Neurology (AAN [25]) recommend an annual cognitive assessment for MS. Regularly monitoring cognition in MS patients can facilitate appropriate management as well as targeted specialist referrals for follow-up expert cognitive assessment and management [26,27]. Once cognitive impairment is identified, healthcare professionals can modify their interaction style with patients and monitor increased risks associated with cognitive impairment such as driving accidents, risk of falls, unemployment and poor disease management [18]. 

In 2012, an international consensus committee of 12 European and American MS experts convened to develop a review process to select scales that could be combined to produce a feasible, valid and international MS cognitive assessment. The committee examined the available cognition scales from the literature, as well as their psychometric qualities and clinical applicability. This approach took account of both the psychometric standards (reliability, validity and sensitivity) and the pragmatic standards (international applicability, ease of administration, patient acceptability and contextual feasibility). The committee agreed that the assessment tool should assess information processing speed, verbal memory and visual memory (immediate recall) and prompted the selection of the following subtests: the Symbol Digit Modalities Test (SDMT; spoken response), the first five learning trials of the California Verbal Learning Test (CVLT-II) and the first three learning trials of the Brief Visuospatial Memory Test-Revised (BVMT-R [28]). These three subtests are reliable and sensitive to MS cognitive impairment.

The SDMT [29] is a measure of information processing speed comprising a key of single numbers, each paired with an abstract symbol. The patient is presented with rows of symbols that are arranged pseudo-randomly. They are required to say the correct number for each of the symbols as fast and as accurately as they can in 90 s, using the key provided. The SDMT shows high sensitivity for MS-related cognitive dysfunction and is now widely acknowledged as the gold standard for a quick cognitive screening [30]. 

In the CVLT-II [31], a measure of verbal memory, only the first five learning trials are administered. The patient is read a 16-item word list at a slightly slower rate than one item per second. The list is read aloud five times, and the patient is instructed to recall as many of the items as possible, in any order, across the five learning trials. 

In the BVMT-R [32], a measure of visual memory, only the first three learning trials are administered. This test involves presenting to patients a 2 × 3 stimulus array of abstract geometric figures across three learning trials, each 10 s in length. The array is then removed from the patient’s view, and they are instructed to draw the geometric figures in the correct position from memory. 

The Brief Cognitive Assessment for Multiple Sclerosis (BICAMS [28]) has been recommended as a 15 min international measure to routinely screen and monitor cognition in MS patients. It was designed for healthcare professionals who may not have specific training in cognitive assessments, allowing more clinics to address cognition. This brief assessment tool does not require any special equipment beyond a pen, paper and stopwatch and therefore allows cognition to be tested inexpensively. BICAMS can be easily implemented into routine clinical practice across centres and countries internationally [28]. The committee have also published an international validation protocol to guide national validation studies [33]. 

BICAMS has been validated in 26 countries to date, including Argentina, Belgium, Turkey and Japan (e.g., [34]). These national studies have investigated the validity and reliability of BICAMS in different cultures and language groups and its sensitivity to cognitive impairment in comparison with the “gold-standard” batteries. The AAN has recommended BICAMS in their quality measurement sets for MS in 2014 and 2020. The Canadian Guidelines for MS Treatment endorsed BICAMS in 2020 [35], and over 20 peer review papers in international clinical neurology journals have also recommended BICAMS for routine cognitive assessment in MS clinics (e.g., [36]). 

BICAMS has been adopted by the international MS community. For example, the Arabic version of BICAMS represents the most used cognitive battery for assessing MS cognition in the Arab world [37]. It has an international reach, with 11,000 patients routinely assessed every year. There has been a systematic review of the first 16 national validation studies on BICAMS [34]. However, there have since been additional national validation studies, warranting an updated systematic review of the validation literature and international findings. The aim of the present systematic review and meta-analysis was to identify, synthesise and critically evaluate current literature on the progress of BICAMS in meeting the objectives of global collaboration and a credible international validation protocol.

## 2. Methods

### 2.1. Search Strategy 

The Preferred Reporting Items for Systematic Reviews and Meta-Analyses (PRISMA) statement was followed as a guide for the standardised conduct and reporting of the current systematic review and meta-analysis [38]. Studies were identified using 3 databases—PubMed, PsycINFO and Web of Science. Boolean search terms were developed and used to identify studies examining the validity of BICAMS in August 2022 (Table 1). Search terms were informed by initial searches and developed further during the process of the review to ensure all relevant articles were identified. 

### 2.2. Selection Criteria

The inclusion criteria were: (a) studies that followed the international validation BICAMS protocol, (b) quantitative studies, (c) peer-reviewed studies with no date restriction that are written in the English language and (d) samples including adults with any clinical subtypes of MS and Clinically Isolated Syndrome (CIS), the MS precursor stage. 

The additional criteria for inclusion in the meta-analysis were as follows: (a) studies including an HC comparison group and (b) studies reporting standard quantitative information based on the SDMT, CVLT-II and BVMT-R subscales (mean, standard deviation and sample size) or appropriate substitute scales of the MS and/or CIS and HC comparison groups. 

### 2.3. Quality Assessment 

One reviewer (HP) extracted data from the studies directly into tables made specifically for the current review, and this was examined and verified by a second reviewer (DL). Two reviewers independently assessed the quality of the retrieved articles using the Effective Public Health Practice Project (EPHPP), and any disagreements were discussed and resolved. A final quality rating was derived from the individual ratings of the categories. 

### 2.4. Statistical Analysis 

The meta-analysis was conducted using the Comprehensive Meta-Analysis (CMA; Version 3) software [39]. Three individual analyses were performed based on the average scores of the SDMT, CVLT-II and BVMT-R subtests for both groups (MS and HC). Effect sizes were calculated as standardised mean differences with Hedges *g* using the following interpretation: 0.2 = small; 0.5 = medium; 0.8 = large [40]. 

The meta-analysis employed a random-effects model because it estimates the mean of a distribution of effects as opposed to one true effect [41,42], and the number of studies are large enough i.e., more than 5 studies. Heterogeneity was assessed using the Cochran’s Q test, and the magnitude of heterogeneity was evaluated using the I^2^ statistic. The I^2^ statistic assesses the percentage of variation across studies that are due to heterogeneity rather than chance and can be interpreted as a small (25%), moderate (50%) or high (75%) level of heterogeneity [43]. 

Forest plots were created for each subtest to visually summarise the amount of heterogeneity as well as the estimated effect sizes (Hedges *g*) and 95% CIs. Funnel plots were also generated as a graphical tool for investigating publication bias and other bias (assessed by the Egger’s test), which, if found, may lead to funnel plot asymmetry [44]. If asymmetry was shown, the Duval and Tweedie trim and fill analysis would model the data as if it were symmetrically distributed by adjusting for missing studies [45].

## 3. Results

### 3.1. Search Results

Using the pre-specified eligibility criteria, 55 results were generated from PubMed, 24 from PsycINFO and 124 from Web of Science. First, 132 duplicate studies across databases were removed (Figure 1). To assess for eligibility, all titles and abstracts were initially screened independently by two reviewers (HP and DL). The 30 full-text articles were re-evaluated to determine their final inclusion or exclusion. Following this, four studies were removed from the final review according to the inclusion criteria. A total of 26 studies met the criteria for final inclusion in the systematic review.

All 26 studies met the criteria for the meta-analysis from those included in the systematic review. All relevant data for the current review and meta-analysis were obtained from numerical information in texts, tables, figures and statistical analysis.

### 3.2. Study Characteristics and Sample Demographics

Data on study characteristics, sample demographics and patient disease information are shown in Table 2. The 26 validation studies were published between the years 2012 and 2022.

Adults with MS were recruited from a variety of settings including medical centres, university hospitals, specialist clinics and tertiary referral centres. HC were either recruited from the community, an established normative sample or among relatives, friends or carers of PwMS. The studies included a total of 2833 adults with MS and 2382 healthy controls. Sample size of both groups differed greatly between studies; in PwMS, the samples ranged from 40 to 500 participants, whilst for HC, this ranged from 20 to 276. Age of PwMS ranged from 20–61 years with an average age of 39.9, whilst the age of HC ranged from 22–51 years, with a similar average age of 38.9. The percentage of females in the MS and HC sample disproportionately favoured females and ranged from 47–82% in the MS sample and 33–86% in the HC. Eight studies used the same number of males and females. Years of education averaged 14.13 years in the MS sample and 14.58 years in HC. Higher rates of employment were seen in the HC in comparison to the MS samples (39–98% compared to 20–89%, respectively).

### 3.3. Patient Disease Information

Six studies recruited an exclusively RRMS sample, whilst the remaining studies also included a mixture of other phenotypes (e.g., SPMS or PPMS). RRMS was the most represented phenotype (33–100%), followed by SPMS (0–38%). Three studies included participants with CIS in their sample. The revised McDonald criteria for MS was the most used diagnostic criterion [72]. The average disease duration was 9.16 years and ranged from 1.08 to 14.67 years. The average Expanded Disability Status Scale (EDSS [73]) score was 2.75, indicating that, on average, the participants were in the mild disability range and could walk unaided.

Few studies calculated sensitivity and specificity data (Table 3), and it is noteworthy that, in the large Czech Republic sample, BICAMS demonstrated the same sensitivity to cognitive impairment as the “gold-standard” Minimal Assessment of Cognitive Function in MS (MACFIMS [52]).

### 3.4. Correlations between BICAMS and Sample Variables

Correlations between BICAMS subtest scores and sample variables (age, disease duration, EDSS score, education, and employment) were extracted (Table 4). Correlations between age and BICAMS scores were the most frequently reported and usually significant; correlations between EDSS scores and BICAMS were occasionally reported and inconsistently significant.

### 3.5. Quality Ratings

The overall quality of the studies ranged from ‘moderate’ to ‘weak’ on the EPHPP template, reflecting the cross-sectional design typical of validation studies. No studies were removed from this review following the quality assessment. 

### 3.6. Meta-Analysis of BICAMS Validation Studies

Data on the standard quantitative information based on the subtests of the SDMT, CVLT-II and BVMT-R of the MS and HC groups were extracted for baseline assessments of BICAMS (Table 3). The percentage of people in both groups identified with likely cognitive impairment on at least one subtest was also extracted, along with the sensitivity and specificity of BICAMS. The results from all three subtests showed that adults with MS performed significantly worse than HC. BICAMS identified likely impaired cognition, on at least one subtest, in 25–73% in the MS sample, which was significantly higher than in HC (1–20%).

The forest plot (Figure 2) shows the effect size for each study using the SDMT. Overall, information processing speed was significantly lower in the MS sample compared to HC with a large effect size (g = 0.854, 95% CI = 0.765, 0.944, *p* < 0.001). There was no evidence of outliers; however, moderate heterogeneity (Q = 51.9, *p* = 0.001) was indicated (I^2^ = 51.8). There was no evidence of publication bias (Egger’s test: *p* > 0.05, two-tailed). The funnel plot (Figure 3) indicates that the effect sizes were symmetrical. Duval and Tweedie’s trim and fill analysis estimated that no studies were missing from the analysis.

A translated version of the CVLT-II was used in 18 validation studies. For two studies, the CVLT-II was not translated as the validation studies were conducted in English-speaking countries with existing validations [62,71]. Importantly, six of the studies used an alternative verbal memory test to substitute the conventional CVLT-II (Table 3). The average mean and standard deviation scores of these alternative tests were included in the meta-analysis. Notably, the study with the smallest effect size, with a Hedge’s *g* value of 0.017, used a substituted verbal memory test ([56]; Figure 4). The study with the highest effect size, with a Hedge’s *g* value of 1.072, used a translated version of the CVLT-II ([57]; Figure 4).

The forest plot (Figure 4) shows the effect size for each study using the CVLT-II. Overall, immediate verbal recall memory was significantly lower in the MS sample compared to HC with a medium effect size (g = 0.566, 95% CI = 0.459, 0.673, *p* < 0.001). There was no evidence of outliers; however, a high level of heterogeneity (Q = 77.9, *p* < 0.001) was indicated (I^2^ = 67.9). Duval and Tweedie’s trim and fill analysis estimated that three studies would need to fall to the left of the mean effect size to make the plot symmetrical (Figure 5). Assuming a random-effects model, the adjusted mean effect size remained medium (*p* = 0.528, 95% CI = 0.420, 0.635). There was no evidence of publication bias, as the Egger’s test remained non-significant (Egger’s test: *p* > 0.05, two-tailed).

The forest plot (Figure 6) shows the effect size for each study using the BVMT-R. Overall, immediate visual recall memory was significantly lower in the MS sample compared to HC with a medium effect size (g = 0.566, 95% CI = 0.487, 0.645, *p* < 0.001). There was no evidence of outliers; however, moderate heterogeneity (Q = 42.6, *p* < 0.05) was indicated (I^2^ = 41.4). There was no evidence of publication bias (Egger’s test: *p* > 0.05, two-tailed). The funnel plot (Figure 7) indicates that the effect sizes were symmetrical. Duval and Tweedie’s trim and fill analysis estimated that no studies were missing from the analysis. 

Only four studies reported the sensitivity and specificity of BICAMS. Of these four studies, one reported on the sensitivity and specificity of BICAMS overall (94% and 86%, respectively), whilst the remaining three reported on the sensitivity and specificity of the individual subtests (see Table 3).

## 4. Discussion

### 4.1. Summary of Findings 

The current review identified, synthesised and appraised the current literature on the international validation of BICAMS to date. A total of 26 studies were included in both the systematic review and meta-analysis. The results from the systematic review showed that BICAMS has been embraced in many countries worldwide and with a range of clinical samples, including different MS phenotypes and consequently, disease durations and severity. Most studies included a HC sample with a similar age and educational background. Although BICAMS was designed to be administered by a range of health professionals, in these validation studies, BICAMS was apparently typically completed by a neuropsychologist or psychology graduate; however, this information was not routinely reported. Finally, in most studies, the gender ratio in both samples disproportionately favoured females. It is important to consider that this female recruitment bias reflects the increased prevalence of MS in females, the female-to-male sex ratio being approximately 3:1 [8]. 

The meta-analysis showed that adults with MS performed significantly worse than HC on the three BICAMS subtests—information processing speed and immediate verbal and visual recall. Cognitive functioning was most impaired on the SDMT (a measure of information processing speed). These findings are in line with existing literature proposing that information processing speed is markedly reduced in MS [74] and constitutes the most common cognitive limitation in PwMS [75]. It is important to stress that BICAMS should be administered in its entirety, given that multiple aspects of daily life can be affected by cognitive impairment in addition to processing speed, e.g., visuospatial learning as assessed by the BVMT-R [76]. 

It is important to note that the BICAMS committee included experts from Europe and America and may lack diversity and inclusivity in development and cross-cultural appropriateness [77,78]. The CVLT-II scores were more heterogeneous compared to the other subtests, possibly reflecting the additional linguistic and cultural demands of translating the verbal recall list. Prior to BICAMS, the CVLT-II had separate word lists and validations for the UK and USA. Six BICAMS validation studies used alternative verbal memory tests available in the required language. Several validation studies [49,51] reported difficulties with translating the CVLT-II and described similar scores on the CVLT-II between the MS sample and HC. The CVLT-II is also probably the most culturally sensitive of the three subtests and required more extensive work to accomplish a valid translation of the stimuli [69]. Semantic categories for the word list were sometimes adapted to be more applicable for the population e.g., by swapping different types of sports for cooking utensils in Egypt [55]. 

### 4.2. Strengths

There are several strengths to this review. First, the search strategy was designed and validated using a combination of three databases—PubMed, PsycINFO and Web of Science—to cover a breadth of the available and relevant literature. Secondly, strict inclusion criteria were employed to ensure appropriate studies were generated. Furthermore, this review identified and synthesised international validation studies reporting objective scores of cognitive abilities in PwMS compared to matched HC in a standardised manner. This review captures the advances in validating BICAMS internationally since the previous review [34], with further validations in 12 more countries. Across the validation studies, there was a varied spread of cultures, languages and countries involved in the initiative. The countries that participated in the international validation protocol reported that BICAMS could be feasibly administered in approximately 15 min, with minimal materials, and was recommended for routine clinical cognitive assessment as a standard of MS care.

### 4.3. Limitations

There are also some notable limitations to the review methodology. First, English-language publication was a requirement for inclusion in the review, so it is important to recognise that this may have limited the inclusion of validation studies published in other languages. Secondly, only the terms “Multiple Sclerosis”, “MS”. “Clinically Isolated Syndrome” or “CIS” were used in the database search. This may have restricted the number of studies identified through the database search, as there are additional ways to describe MS (e.g., as an autoimmune disease). Thirdly, as part of the pre-defined criteria, only peer-reviewed studies were considered eligible for inclusion in this review, which meant that possible grey literature (e.g., thesis publications) that were not commercially published would not have been included. Fourthly, there are likely to be international disparities across studies in relation to healthcare systems, accessibility, economic status, and access to general MS support facilities [79,80]. MS healthcare in countries with developing economies may be constrained by limited access to high-efficacy disease-modifying therapies (DMTs) or diagnostic technology such as magnetic resonance imaging (MRI [81]). Developed countries have significantly higher prevalence and incidence rates of MS compared to developing countries, which may reflect better access to diagnostic facilities and subsequent earlier diagnosis and treatment [82]. These variations in access and quality of MS healthcare may have made comparisons of disease profiles, such as years since diagnosis and physical disability, less valid. Most of the studies included in this review were conducted in leading centres and university hospitals, which attract a certain sociodemographic population and, therefore, may not be entirely representative of all MS populations. Fifthly, there was a great deal of heterogeneity between studies—namely in terms of sample size, age, MS phenotypes and disease duration. RRMS was overrepresented compared to other MS phenotypes. It is possible that this may have reduced the effect size since cognitive impairment is more common and severe in the progressive forms of the disease [10,11]. With progressive forms of MS being underrepresented in this review, cognitive impairment may also have been underrepresented in the identified studies compared to the general MS population. Finally, the quality assessment tool (EPHPP) used to analyse the methodological quality of the included studies may not have been considered appropriate in this systematic review, since it is not a scale designed for cross-sectional studies. This may explain why the overall quality of the studies ranged from ‘moderate’ to ‘weak’ on the EPHPP template. In addition, the possible risk of bias was not studied.

### 4.4. Future Directions 

The adoption of an international validation protocol and a global collaboration have served to promote BICAMS to international currency for MS cognition. This is reflected in the number of international validations published, the report of BICAMS data in 150 published studies of MS cognition and its use in many large national and international trials. This initiative could serve as a model for other conditions, improving the awareness, understanding, assessment and management of cognitive impairment. It is hoped that further research investigating the feasibility of BICAMS in clinical practice will maximise its use in routine consultation to evaluate cognitive status in MS. This systematic review also prompts future studies to investigate the sensitivity and specificity of the scale in different forms of multiple sclerosis or in groups with different degrees of disability.

## 5. Conclusions

BICAMS has been translated and culturally adapted in 26 countries to date. It has been shown to be a valid measure of cognitive functioning in MS at a global level. It can detect cognitive impairment in individuals with MS compared to healthy controls across a range of cultures, languages, and countries. This review sheds light on the work of the international MS community at validating BICAMS utilising an international validation protocol. This represents progress in the increasing awareness of MS cognition as well as maximising the implementation of BICAMS into routine clinical practice, to assess and instigate the appropriate management of MS cognition across different countries.

## Figures and Tables

**Figure 1 jcm-12-00703-f001:**
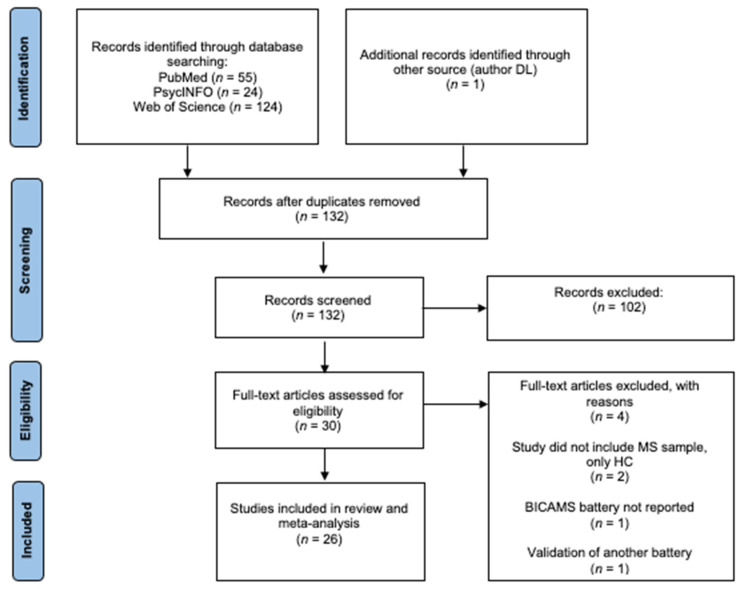
PRISMA flowchart for selection process of studies in systematic review and meta-analysis.

**Figure 2 jcm-12-00703-f002:**
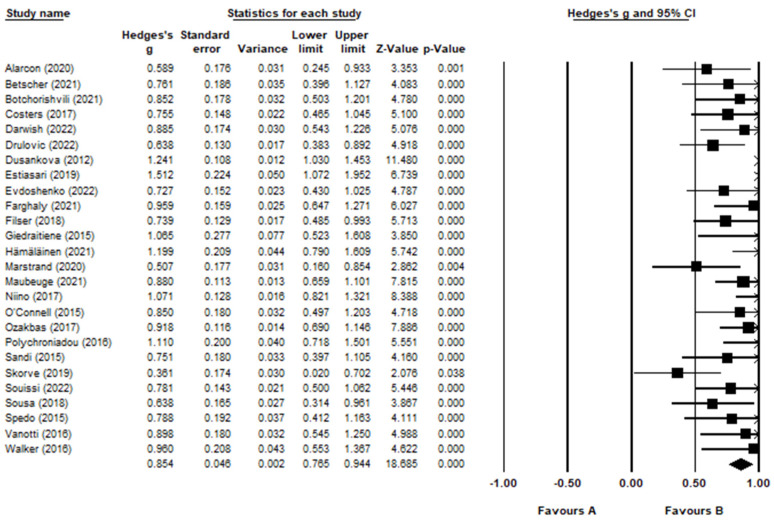
Forest plot for SDMT. Alarcón et al. [46]; Betscher et al. [47]; Botchorishvili et al. [48]; Costers et al. [49]; Darwish et al. [50]; Drulović et al. [51]; Dusankova et al. [52]; Estiasari et al. [53]; Evdoshenko et al. [54]; Farghaly et al. [55]; Filser et al. [56]; Giedraitienė et al. [57]; Hämäläinen et al. [58]; Marstrand et al. [59]; Maubeuge et al. [60]; Niino et al. [61]; O’Connell et al. [62]; Ozakbas et al. [63]; Polychroniadou et al. [64]; Sandi et al. [65]; Skorve et al. [66]; Souissi et al. [67]; Sousa et al. [68]; Spedo et al. [69]; Vanotti et al. [70]; Walker et al. [71].

**Figure 3 jcm-12-00703-f003:**
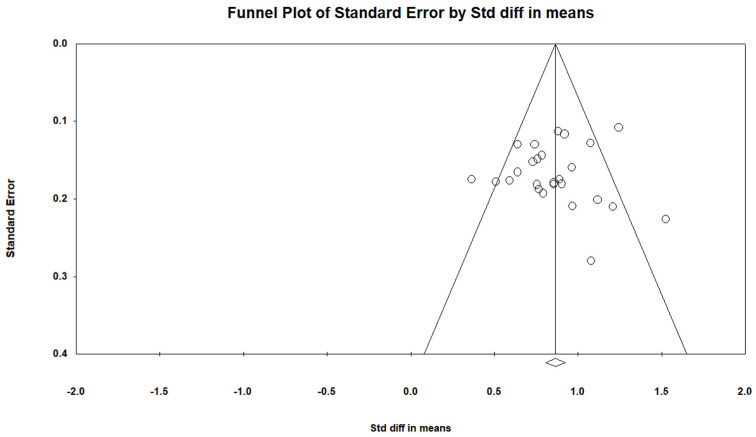
Funnel plot for SDMT.

**Figure 4 jcm-12-00703-f004:**
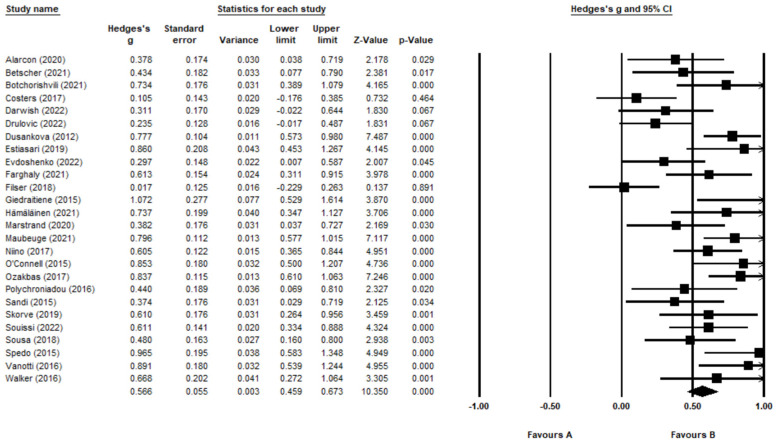
Forest Plot for CVLT-II. Alarcón et al. [46]; Betscher et al. [47]; Botchorishvili et al. [48]; Costers et al. [49]; Darwish et al. [50]; Drulović et al. [51]; Dusankova et al. [52]; Estiasari et al. [53]; Evdoshenko et al. [54]; Farghaly et al. [55]; Filser et al. [56]; Giedraitienė et al. [57]; Hämäläinen et al. [58]; Marstrand et al. [59]; Maubeuge et al. [60]; Niino et al. [61]; O’Connell et al. [62]; Ozakbas et al. [63]; Polychroniadou et al. [64]; Sandi et al. [65]; Skorve et al. [66]; Souissi et al. [67]; Sousa et al. [68]; Spedo et al. [69]; Vanotti et al. [70]; Walker et al. [71].

**Figure 5 jcm-12-00703-f005:**
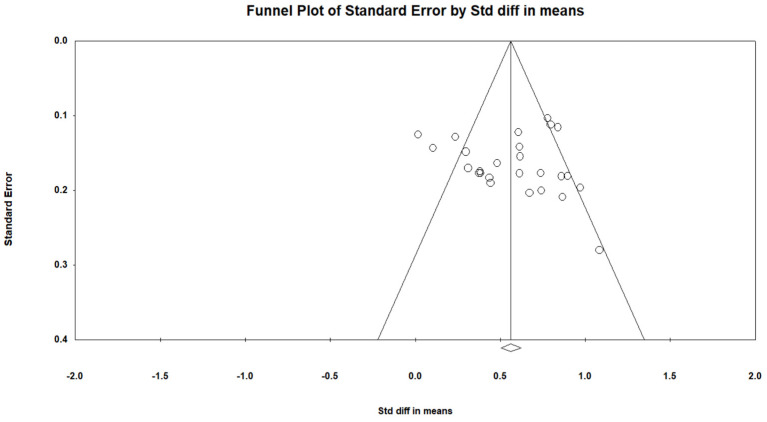
Funnel plot for CVLT-II.

**Figure 6 jcm-12-00703-f006:**
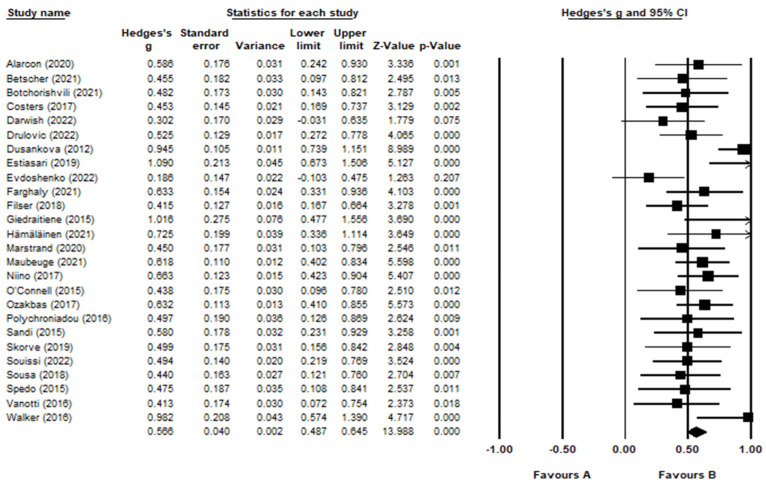
Forest plot for BVMT-R. Alarcón et al. [46]; Betscher et al. [47]; Botchorishvili et al. [48]; Costers et al. [49]; Darwish et al. [50]; Drulović et al. [51]; Dusankova et al. [52]; Estiasari et al. [53]; Evdoshenko et al. [54]; Farghaly et al. [55]; Filser et al. [56]; Giedraitienė et al. [57]; Hämäläinen et al. [58]; Marstrand et al. [59]; Maubeuge et al. [60]; Niino et al. [61]; O’Connell et al. [62]; Ozakbas et al. [63]; Polychroniadou et al. [64]; Sandi et al. [65]; Skorve et al. [66]; Souissi et al. [67]; Sousa et al. [68]; Spedo et al. [69]; Vanotti et al. [70]; Walker et al. [71].

**Figure 7 jcm-12-00703-f007:**
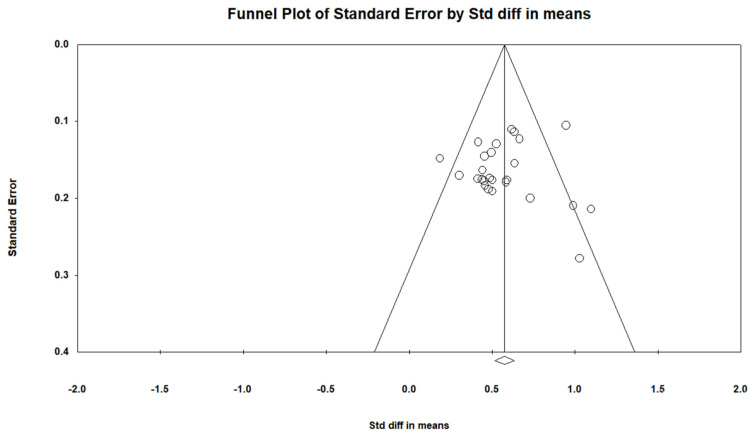
Funnel plot for BVMT-R.

**Table 1 jcm-12-00703-t001:** Search terms for systematic review.

Search Terms
“Multiple Sclerosis” OR “MS” OR “Clinically Isolated Syndrome” OR “CIS” AND “Brief International Cognitive Assessment for Multiple Sclerosis” OR “BICAMS” AND “Validation” OR “International Validation” OR “Validity” OR “Sensitivity”

**Table 2 jcm-12-00703-t002:** Study characteristics and sample demographic and patient disease information.

Study	Country	Number of Participants	Age in Years Mean (SD), {Median}, [Range]	Gender (Female %)	Education in Years Mean (SD), {Median}, [Range]	Employment (Employed %)	MS Phenotype %(CIS/RR/SP/PP/PR)	Disease Duration in Years Mean (SD), {Median}, [Range]	EDSS Mean (SD), {Median}, [Range]
Alarcón et al. [46]									
MS	Columbia	50	41.44 (10.99)	64%	14.76 (2.61)	Nr	0/100/0/0/0	7.66 (5.61)	1.33 (1.54)
HC		100	37.75 (12.63)	48%	14.73 (3.57)	Nr	-	-	-
Betscher et al. [47]									
MS	Poland	61	{39}	74%	{13}	84%	0/74/20/6/0	RR = {5} SP = {19.5} PP = {7.5}	RR = {3} SP = {4.75} PP = {4.5}
HC		61	{37}	75%	{13}	98%	-	-	-
Botchorishvili et al. [48]									
MS	Georgia	68	39.2 (9.9)	71%	14.3 (2.1)	57%	0/76/18/6/0	7.0 (5.7)	3.3 (1.6)
HC		68	38.5 (9.9)	68%	14.5 (1.9)	84%	-	-	-
Costers et al. [49]									
MS	Belgium	97	45.42 (9.24)	68%	14.28 (1.86)	Nr	0/84/12/4/0	12.97 (7.16)	3.50 (2.50)
HC		97	43.52 (12.69)	75%	14.69 (1.61)	Nr	-	-	-
Darwish et al. [50]									
MS	Lebanon	43	36.06 (12.37)	81.4%	14.63 (3.17)	48.84%	0/81/14/5/0	8.61 (7.36)	1.89 (1.7)
HC		180	45.01 (19.36)	60%	15.13 (3)	56.11%	-	-	-
Drulović et al. [51]									
MS	Serbia	500	39.9 (9.4)	70.2%	14.0 (2.9)	Nr	0/100/0/0/0	9.2 (6.7)	{2.0}
HC		69	40.3 (11.5)	63.77%	14.1 (3.4)	Nr	-	-	-
Dusankova et al. [52]									
MS	Czech Republic	367	34 (10)	68%	14 (3)	40%	0/68/26/3/3	8 (7)	3 (1.5)
HC		134	34 (9)	71%	14 (2.5)	73%	-	-	-
Estiasari et al. [53]									
MS	Indonesia	40	{31}, [20–61]	82.5%	>12 yrs = 75%	Nr	0/78/22/0/0	{4}, [0.1–15]	{3}, [1–7.5]
HC		66	{29}, [22–51]	72.7%	>12 yrs = 89.4%	Nr	-	-	-
Evdoshenko et al. [54]									
MS	Russia	98	38.44 (11.47)	70.4%	15.12 (2.79)	Nr	0/86/14/0/0	9.5 (7.44)	{3.0}
HC		86	38.17 (13.29)	63.95%	16.26 (3.02)	Nr	-	-	-
Farghaly et al. [55]									
MS	Egypt	90	30.8 (6.7)	77.78%	14.5 (2.6)	Nr	0/86/12/2/0	6.2 (5.8)	2.8 (1.8)
HC		85	30.5 (7.9)	70.59%	14.3 (3.3)	Nr	-	-	-
Filser et al. [56]									
MS	Germany	172	43.33 (11.64)	68%	10.74 (1.56)	76.4%	0/87/9/4/0	Nr	Nr
HC		100	43.04 (15.59)	71%	10.77 (1.58)	92%	-	-	-
Giedraitienė et al. [57]									
MS	Lithuania	50	38.8 (10.2)	47%	15.9 (2.8)	54%	4/88/6/2/0	11.7 (9.2)	3.3 (1.3)
HC		20	36.7 (16.4)	33%	17.5 (3.5)	75%	-	-	-
Hämäläinen et al. [58]									
MS	Finland	65	50.9 (8.8)	71%	13.8 (9.8)	20%	0/62/38/0/0	15.9 (9.8)	4.8 (2.0)
HC		45	49.4 (12.6)	71%	14.0 (2.1)	86.7%	-	-	-
Marstrand et al. [59]									
MS	Denmark	65	37.2 (8.8)	63%	15.2 (2.4)	Nr	0/100/0/0/0	3.9 (2.7)	1.8 (1.2)
HC		65	36.8 (9.6)	63%	15.9 (2.1)	Nr	-	-	-
Maubeuge et al. [60]									
MS	France	123	49.69 (9.41)	63.4%	14–16 yrs = 30.1%	44.7%	0/33/33/34/0	14.67 (9.09)	{4.0}, [0–8]
HC		276	43.84 (12.42)	57.3%	14–16 yrs = 38%	Nr	-	-	-
Niino et al. [61]									
MS	Japan	156	41.4 (9.3)	69%	14.1 (1.9)	Nr	0/88/11/1/0	10.3 (7.2)	2.4 (2.0)
HC		126	39.3 (11.9)	72%	14.3 (1.6)	Nr	-	-	-
O’Connell et al. [62]									
MS	Ireland	67	42.7 (12.8)	68%	14.1 (3.1)	41.8%	0/70/28/2/0	10.2 (8.4)	1.8 (0.9)
HC		66	43.9 (12.1)	73%	13.6 (2.7)	80.3%	-	-	-
Ozakbas et al. [63]									
MS	Turkey	173	37.5 (10.7)	71%	13.9 (7.3)	23.7%	0/87/10/3/0	9.2 (6.1)	2.4 (1.7)
HC		153	36.9 (8.9)	71%	15.4 (8.8)	39.1%	-	-	-
Polychroniadou et al. [64]									
MS	Greece	44	40.2 (9.9)	61%	13.9 (4.2)	Nr	7/77/9/7/0	9.1 (4.1)	{3.5}, [1.0–6.0]
HC		79	36.2 (10.6)	60%	15.6 (5.5)	Nr	-	-	-
Sandi et al. [65]									
MS	Hungary	65	41.9 (8.9)	75%	>12 yrs = 52.3%	Nr	0/100/0/0/0	11.1 (7.6)	2.5 (1.8)
HC		65	40.9 (11.8)	75%	>12 yrs = 52.3%	Nr	-	-	-
Skorve et al. [66]									
MS	Norway	65	37.02 (0.40)	64.6%	14–16 yrs = 37%	89.2%	0/100/0/0/0	1.08 (0.74)	1.28 (0.88)
HC		68	38.13 (11.40)	66.2%	14–16 yrs = 46%	97.0%	-	-	-
Souissi et al. [67]									
MS	Tunisia	104	33.3 (9.8)	75%	14–16 yrs = 14.42%	Nr	0/88/8/4/0	7 (6.4)	2.65 (2.06)
HC		104	33.3 (9.4)	75%	14–16 yrs = 14.42%	Nr	-	-	-
Sousa et al. [68]									
MS	Portugal	105	38.26 (11.03)	66.7%	13.55 (3.71)	58.1%	4/92/4/0/0	6.52 (5.95)	{1.5}, [0–6]
HC		60	36.17 (12.01)	58.3%	14.62 (3.47)	94.9%	-	-	-
Spedo et al. [69]									
MS	Brazil	58	41.2 (12.2)	69%	12.7 (5.2)	Nr	0/100/0/0/0	8.3 (6.6)	4.2 (2)
HC		58	40.3 (11.9)	55%	12.5 (3.6)	Nr	-	-	-
Vanotti et al. [70]									
MS	Argentina	50	43.4 (10.2)	74%	14.9 (2.8)	Nr	0/78/18/4/0	13.1 (9.1)	3.29 (2.55)
HC		100	42.4 (10.1)	75%	14.9 (2.5)	Nr	-	-	-
Walker et al. [71]									
MS	Canada	57	45.4 (9.9)	80%	15.44 (2.7)	Nr	0/77/16/7/0	10.11 (7.72)	2.7 (1.85)
HC		51	41.9 (10.8)	86%	16.31 (2.1)	Nr	-	-	-

MS = multiple sclerosis; HC = healthy control; CIS = clinically isolated syndrome; RR = relapsing–remitting MS; SP = secondary progressive MS; PP = primary progressive MS; PR = progressive–relapsing MS; EDSS = expanded disability status scale; Nr = not reported; SD = standard deviation.

**Table 3 jcm-12-00703-t003:** BICAMS psychometrics.

Study	SDMT Score Mean (SD)	CVLT-II Score Mean (SD)	BVMT-R SCORE Mean (SD)	Impaired Cognition on at Least One Subtest (%)	Sensitivity (%)	Specificity (%)
Alarcón et al. [46]						
MS	46.47 (14.24)	45.34 (10.14) ^a^	21.64 (6.91)	50%	Nr	Nr
HC	54.11 (12.19)	48.78 (8.45) ^a^	25.67 (6.81)	-	-	-
Betscher et al. [47]						
MS	48.8 (12.1)	51.7 (10.9)	24 (7.7)	34%	Nr	Nr
HC	57.2 (9.7)	56.1 (9.2)	27.1 (5.7)	Nr	-	-
Botchorishvili et al. [48]						
MS	35.5 (12.7)	51.0 (11.8)	22.0 (8.0)	43%	Nr	Nr
HC	46.0 (11.8)	58.5 (8.2)	25.6 (6.8)	14%	-	-
Costers et al. [49]						
MS	52.1 (13.1)	60.1 (12.9)	25.4 (29)	Nr	Nr	Nr
HC	61 (10.2)	61.3 (9.7)	28.2 (5.1)	Nr	-	-
Darwish et al. [50]						
MS	47.2 (17.98)	56.9 (10.04) ^b^	22 (9.79)	61%	Nr	Nr
HC	59.22 (12.27)	54.10 (8.71) ^b^	24.23 (6.66)	Nr	-	-
Drulović et al. [51]						
MS	45.9 (16.7)	50.0 (11.7)	18.8 (7.4)	62.9%	Nr	Nr
HC	56.3 (12.9)	52.7 (9.6)	22.6 (5.8)	18.6%	-	-
Dusankova et al. [52]						
MS	50 (13)	52 (11)	23 (7)	58%	94%	86%
HC	65 (9)	60 (8)	29 (4)	0.7%	-	-
Estiasari et al. [53]						
MS	40.9 (14.8)	52.0 (12.8)	22.2 (7.7)	40%	Nr	Nr
HC	64.8 (16.2)	61.5 (9.7)	29.3 (5.6)	Nr	-	-
Evdoshenko et al. [54]						
MS	49.16 (13.42)	{61.5}	{26.5}	34.69%	Nr	Nr
HC	58.34 (11.52)	{65.5}	{28}	16.28%	-	-
Farghaly et al. [55]						
MS	39.2 (13.3)	53.7 (10.5)	19.7 (9.2)	SDMT = 31.1% CVLT-II = 19.5% BVMT-R = 23.9%	Nr	Nr
HC	50.9 (10.8)	59.6 (8.5)	25.4 (8.7)	SDMT = 5.8% CVLT-II = 7% BVMT-R = 8.1%	-	-
Filser et al. [56]						
MS	47.43 (11.67)	55.35 (11.43) ^c^	24.44 (7.59)	32.6%	Nr	Nr
HC	56.07 (11.64)	55.16 (10.27) ^c^	27.37 (5.96)	Nr	-	-
Giedraitienė et al. [57]						
MS	42.7 (13.9)	55.9 (10)	23.1 (7)	Nr	Nr	Nr
HC	57 (11.5)	65.7 (5.9)	29.6 (4.1)	Nr	-	-
Hämäläinen et al. [58]						
MS	41.9 (11.8)	43.0 (11.5)	19.2 (8.0)	60%	Nr	Nr
HC	54.6 (8.3)	51.3 (10.7)	24.7 (6.8)	Nr	-	-
Marstrand et al. [59]						
MS	61.0 (10.0)	65.4 (9.9)	27.4 (5.8)	32.3%	SDMT = 20.0% CVLT-II = 10.8% BVMT-R = 16.9%	SDMT = 95.4% CVLT-II = 89.2% BVMT-R = 93.8%
HC	66.0 (9.6)	68.6 (6.4)	29.6 (3.7)	20%	-	-
Maubeuge et al. [60]						
MS	50.31 (11.12)	49.72 (12.77) ^d^	22.89 (7.26)	50.4%	Nr	Nr
HC	58.55 (8.44)	57.78 (8.67) ^d^	26.73 (5.67)	19.6%	-	-
Niino et al. [61]						
MS	47.9 (14)	48.6 (12.6)	23.5 (8.4)	Nr	Nr	Nr
HC	61 (9.5)	55.7 (10.5)	28.3 (5.4)	Nr	-	-
O’Connell et al. [62]						
MS	46.0 (12.9)	45.3 (10.2)	17.9 (7.1)	57%	Nr	Nr
HC	56.1 (10.6)	53.6 (9.1)	20.9 (6.5)	17%	-	-
Ozakbas et al. [63]						
MS	43.2 (12.5)	45.7 (11.3)	16.9 (8.5)	45.1%	Nr	Nr
HC	53.5 (9.5)	53.9 (7.7)	22.5 (9.2)	Nr	-	-
Polychroniadou et al. [64]						
MS	45.0 (17.2)	55.5 (12.3) ^e^	18.5 (8.3)	47%	Nr	Nr
HC	61.4 (13.1)	60.5 (10.7) ^e^	22.1 (6.5)	Nr	-	-
Sandi et al. [65]						
MS	55.6 (15.5)	55.4 (10.7)	22.5 (8.5)	52.3%	Nr	Nr
HC	66.2 (12.4)	59.0 (8.3)	26.7 (5.6)	Nr	-	-
Skorve et al. [66]						
MS	54.65 (10.79)	54.55 (10.86)	26.55 (5.76)	46.2%	Nr	Nr
HC	58.52 (10.53)	60.32 (7.75)	29.03 (4.01)	Nr	-	-
Souissi et al. [67]						
MS	36 (13)	42 (7) ^f^	23 (9)	73.1%	SDMT = 74% TVLT = 76% ^f^ BVMT-R = 75%	SDMT = 56% TVLT = 55% ^f^ BVMT-R = 53.5%
HC	47 (15)	46 (6) ^f^	27 (7)	Nr	-	-
Sousa et al. [68]						
MS	51.77 (11.20)	55.05 (11.84)	21.72 (7.27)	24.8%	Nr	Nr
HC	58.68 (10.02)	60.47 (10.12)	24.68 (5.52)	Nr	-	-
Spedo et al. [69]						
MS	35.9 (16.1)	42.1 (12.4)	19.9 (8.6)	Nr	Nr	Nr
HC	47.5 (13)	53.4 (10.8)	23.8 (7.7)	Nr	-	-
Vanotti et al. [70]						
MS	45.1 (16.1)	50.9 (12.4)	20.7 (7.74)	Nr	Nr	Nr
HC	56.7 (10.9)	60.9 (10.5)	23.4 (5.8)	Nr	-	-
Walker et al. [71]						
MS	49.7 (10.8)	51.6 (10.1)	24.6 (6.5)	57.9%	SDMT = 97.5% CVLT-II = 82.5% BVMT-R = 77.5%	SDMT = 88.2% CVLT-II = 70.6% BVMT-R = 82.4%
HC	59.1 (8.5)	57.7 (7.9)	29.8 (3.6)	Nr	-	-

MS = multiple sclerosis; HC = healthy control; SDMT = symbol digit modalities test; CVLT-II = California verbal learning test; BVMT-R = brief visuospatial memory test-revised; Nr = not reported; SD = standard deviation. ^a^ Alternative verbal memory test used = The Prueba de Aprendizaje y Memoria con Codificación Libre (PAMCL). ^b^ Alternative verbal memory test used = The Verbal Memory Arabic Test (VMAT). ^c^ Alternative verbal memory test used = The Rey Auditory Verbal Learning Test (RAVLT). ^d^ Alternative verbal memory test used = The French Verbal Learning Test (FVLT). ^e^ Alternative verbal memory test used = The Greek Verbal Learning Test (GVLT). ^f^ Alternative verbal memory test used = The Tunisian Verbal Learning Test (TVLT).

**Table 4 jcm-12-00703-t004:** Correlations between BICAMS scores and MS sample variables.

Study	BICAMS Scores and Sample Variables											
	Age *(r)*			Disease Duration *(r)*			EDSS *(r)*			Education Years *(r)*		
	SDMT	CVLT-II	BVMT-R	SDMT	CVLT-II	BVMT-R	SDMT	CVLT-II	BVMT-R	SDMT	CVLT-II	BVMT-R
Alarcón et al. [46]												
MS	Nr	-	Nr	Nr	-	Nr	Nr	-	Nr	Nr	-	Nr
HC	Nr	-	Nr	-	-	-	-	-	-	Nr	-	Nr
Betscher et al. [47]												
MS	−0.28 *	Nr	−0.26 *	Nr	Nr	Nr	−0.58 ***	−0.31 *	−0.27 *	0.36 *	0.42 ***	0.5 ***
HC	−0.35 *	Nr	Nr	-	-	-	-	-	-	0.44 ***	0.47 ***	0.27 *
Botchorishvili et al. [48]												
MS	−0.400*	−0.112	−0.192	−0.177	−0.106	0.125	−0.582 ***	−0.403 ***	−0.342 ***	0.243 *	0.207	0.297 *
HC	−0.457 ***	−0.368 ***	−0.506 ***	-	-	-	-	-	-	0.523 ***	0.439 *	0.348 *
Costers et al. [49]												
MS	−0.34 ***	−0.10	−0.29 **	Nr	Nr	Nr	−0.44 ***	−0.35 ***	−0.43 ***	Nr	Nr	Nr
HC	Nr	Nr	Nr	-	-	-	-	-	-	Nr	Nr	Nr
Darwish et al. [50]												
MS	Nr	-	Nr	Nr	-	Nr	Nr	-	Nr	Nr	-	Nr
HC	Nr	-	Nr	-	-	-	-	-	-	Nr	-	Nr
Drulović et al. [51]												
MS	−0.225 *	−0.232 ***	−0.271 ***	−0.109 ***	−0.880	−0.207 ***	−0.466 ***	−0.320 ***	−0.360 ***	0.339 ***	0.298 ***	0.190 ***
HC	−0.605 ***	−0.430 ***	−0.374 ***	-	-	-	-	-	-	0.521 ***	0.552 ***	0.394 ***
Dusankova et al. [52]												
MS	Nr	Nr	Nr	0.44 ***	0.39 ***	0.41 ***	Nr	Nr	Nr	Nr	Nr	Nr
HC	Nr	Nr	Nr	-	-	-	-	-	-	Nr	Nr	Nr
Estiasari et al. [53]												
MS	−0.004	−0.11	0.02	−0.23	−0.19	−0.18	−0.5 *****	−0.46 ***	−0.49 ***	{47}, [15–69]	{54}, [23–72]	{24.5}, [4–32]
HC	−0.27 ***	−0.11	−0.28 ***	-	-	-	-	-	-	{63}, [42–110] *	{63}, [36–77]	{31}, [14–36]
Evdoshenko et al. [54]												
MS	Nr	Nr	Nr	Nr	Nr	Nr	Nr	Nr	Nr	Nr	Nr	Nr
HC	Nr	Nr	Nr	-	-	-	-	-	-	Nr	Nr	Nr
Farghaly et al. [55]												
MS	−0.26 ^a,^***	−0.17 ^a^	−0.26 ^a,^***	−0.41 ^a,^*****	−0.18 ^a^	−0.27 ^a,^***	−0.37 ^a,^*****	−0.31 ^a,^***	−0.19 ^a^	0.36 ^a,^*****	0.27 ^a,^***	0.25 ^a,^***
HC	Nr	Nr	Nr	-	-	-	-	-	-	Nr	Nr	Nr
Filser et al. [56]												
MS	Nr	-	Nr	Nr	-	Nr	Nr	-	Nr	Nr	-	Nr
HC	Nr	-	Nr	-	-	-	-	-	-	Nr	-	Nr
Giedraitienė et al. [57]												
MS	Nr	Nr	Nr	−0.3 ^a^	−0.2 ^a^	−0.2 ^a^	−5.9 ^a,^*****	−3.7 ^a,^*****	−2.3 ^a,^*****	2.4 ^a,^***	2.4 ^a,^***	1.0 ^a,^***
HC	Nr	Nr	Nr	-	-	-	-	-	-	2.0 ^a,^***	1.2 ^a,^***	0.9 ^a,^***
Hämäläinen et al. [58]												
MS	Nr	Nr	Nr	Nr	Nr	Nr	Nr	Nr	Nr	Nr	Nr	Nr
HC	Nr	Nr	Nr	-	-	-	-	-	-	Nr	Nr	Nr
Marstrand et al. [59]												
MS	Nr	Nr	Nr	Nr	Nr	Nr	Nr	Nr	Nr	Nr	Nr	Nr
HC	Nr	Nr	Nr	-	-	-	-	-	-	Nr	Nr	Nr
Maubeuge et al. [60]												
MS	Nr	-	Nr	Nr	-	Nr	Nr	-	Nr	Nr	-	Nr
HC	Nr	-	Nr	-	-	-	-	-	-	Nr	-	Nr
Niino et al. [61]												
MS	–0.37 *****	–0.25 ***	–0.30 ***	–0.30 ***	–0.12	–0.27 ***	–0.56 *****	–0.29 *****	−0.46 *****	0.07	0.13	0.001
HC	–0.44 *****	–0.23 ***	–0.25 ***	-	-	-	-	-	-	0.24 ***	0.25 ***	0.05
O’Connell et al. [62]												
MS	Nr	Nr	Nr	Nr	Nr	Nr	Nr	Nr	Nr	Nr	Nr	Nr
HC	Nr	Nr	Nr	-	-	-	-	-	-	Nr	Nr	Nr
Ozakbas et al. [63]												
MS	Nr	Nr	Nr	Nr	Nr	Nr	−0.46 ***	−0.40 ***	−0.24	Nr	Nr	Nr
HC	Nr	Nr	Nr	-	-	-	-	-	-	Nr	Nr	Nr
Polychroniadou et al. [64]												
MS	Nr	Nr	Nr	Nr	Nr	Nr	Nr	Nr	Nr	Nr	Nr	Nr
HC	Nr	Nr	Nr	-	-	-	-	-	-	Nr	Nr	Nr
Sandi et al. [65]												
MS	Nr	Nr	Nr	Nr	Nr	Nr	Nr	Nr	Nr	Nr	Nr	Nr
HC	Nr	Nr	Nr	-	-	-	-	-	Nr	Nr	Nr	Nr
Skorve et al. [66]												
MS	Nr	Nr	Nr	Nr	Nr	Nr	Nr	Nr	Nr	Nr	Nr	Nr
HC	Nr	Nr	Nr	-	-	-	-	-	-	Nr	Nr	Nr
Souissi et al. [67]												
MS	Nr	-	Nr	Nr	-	Nr	Nr	-	Nr	Nr	-	Nr
HC	Nr	-	Nr	-	-	-	-	-	-	Nr	-	Nr
Sousa et al. [68]												
MS	Nr	Nr	Nr	Nr	Nr	Nr	−0.497 *****	−0.334 *****	−0.275 ***	Nr	Nr	Nr
HC	Nr	Nr	Nr	-	-	-	-	-	-	Nr	Nr	Nr
Spedo et al. [69]												
MS	−0.30 ***	−0.30 ***	−0.29 ***	Nr	Nr	Nr	Nr	Nr	Nr	0.29 ***	0.18 ***	0.27 ***
HC	−0.49 ***	-	−0.34 ***	-	-	-	-	-	-	0.49 ***	0.37 ***	-
Vanotti et al. [70]												
MS	Nr	Nr	Nr	Nr	Nr	Nr	Nr	Nr	Nr	Nr	Nr	Nr
HC	Nr	Nr	Nr	-	-	-	-	-	-	Nr	Nr	Nr
Walker et al. [71]												
MS	Nr	Nr	Nr	Nr	Nr	Nr	Nr	Nr	Nr	-	0.20 ***	-
HC	Nr	Nr	Nr	-	-	-	-	-	-	Nr	Nr	Nr

MS = multiple sclerosis; HC = healthy control; SDMT = symbol digit modalities test; CVLT-I I= California verbal learning test; BVMT-R = brief visuospatial memory test-revised; EDSS = expanded disability status scale; Nr = not reported. ^a^ Regression coefficient reported; correlation coefficients (r) are presented with significance marks: * *p* < 0.05, ** *p* < 0.01, *** *p* < 0.001.

## Data Availability

All data analysed and reported in this review are included in this published article.

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
