# Peer review of "A Systematic Review and Meta-Analysis of the Brief Cognitive Assessment for Multiple Sclerosis (BICAMS) International Validations"

_jcm, 2023, doi:10.3390/jcm12020703_

Round 1

Reviewer 1 Report

The authors discuss the use of the BICAMS scale as a valid tool for the assessment of cognitive status in patients with multiple sclerosis, due to the versatility of the scale and its quick and easy administration. However, there are several aspects throughout the paper that should be improved for better understanding.

One element that should be corrected is the abbreviations in the entire manuscript, including figures and tables. There are abbreviations that do not have the complete form, such as SDMT, CVLT-II and BVMT-R in lines 98 and 99; BRB-N and MACFIMS in line 117; HC in line 164 or DMT in line 477. In addition, the abbreviations used in each figure and table should be included in its footnote and not only in some of them. In Table 3 there are abbreviations repeated in the footnote in two different forms, one of which is not used.

Introduction

The introductory section is too long and extensive. I would reduce the part on the importance of cognitive assessment, as it is too much information, and this subject is overexplained. Lines 39, 40 and 41, which explain what was done in the study, it should go at the end next to the objective.

Methods.

The tool used to analyze methodological quality was not appropriate, since it is not a scale designed for cross-sectional studies. In addition, the possible risk of bias has not been studied.

Results

The process followed for the selection of the studies is not clear. It is not described how many studies were selected after reading the title and abstract; therefore, it is not known how many were read in full text. The numbers in the flow chart do not match at all with what is described in the text.

Information in lines 202 and 203 of studies eliminated for not having people with multiple sclerosis is not necessary, since it is described in the inclusion criteria.

There are six studies with alternative verbal memory test. Do these tests have the same items evaluated, and do they have the same score for inclusion in the meta-analysis? Due to the large number of studies analyzed, would it not be better to eliminate these studies from the meta-analysis?

It would be advisable to include a title for each forest plot, indicating in the image itself what it is A and B, as well as including the heterogeneity values, and the total value of p and z.

Maybe it is not possible because of the way the data are presented in the studies. But it would be interesting to divide the meta-analysis study into the remitting and progressive forms of multiple sclerosis to see the possible differences.

Discussion

The discussion is well-structured, easy to follow and understand, and well-supported. However, I would find it interesting to further investigate the sensitivity and specificity of the scale in different forms of multiple sclerosis or in groups with different degrees of disability.

In the conclusions section, I would try to focus more on the results obtained in the study itself.

Reviewer 2 Report

The manuscript submitted by Potticary and Langdon presents a systematic review and meta-analysis of the international validation of BICAMS. The paper is well written and the discussion section is well organized and discusses the current findings with those previously published. The article is important for the sclerosis literature. It deals with the extremely important problem of cognitive dysfunction in patients with multiple sclerosis, and discusses the role of the use of neuropsychological tests (BICAMS battery) in the diagnosis of CI, based on 26 national validations. The data from the above publication are very important because they confirm the prevalence of CI, they indicate the main domains subject to disturbance, what is more, regardless of the place of the study.

My comments on further improving the manuscript are as follows:

Verse 42 - It says multiple sclerosis is a neurodegenerative disease. I can't quite agree with that. MS is an inflammatory-neurodegenerative disease, the first phase is dominated by the inflammatory phase, the second, mainly in progressive forms, neurodegenerative. I would suggest changing that.

It seems to me that the number of studies that have been analyzed is not consistent with what is shown in Figure 1. In the text, we have that 26 studies were subject to final evaluation, and in the figure, the final number is 13. Other data on the analysis of databases with what is shown in Figure 1 also differ. Pages 5 and 6 of the paper.

I think that for greater clarity of the article in Figures 2 and 3, it should be clearly marked which parts concerning SDMT, CLVT-II and BVMT-R
